# Global musical diversity is largely independent of linguistic and genetic histories

Sam Passmore [1,2] ✉, Anna L. C. Wood[3] ✉, Chiara Barbieri [4,5,6], Dor Shilton[7,8], Hideo Daikoku[1], Quentin D. Atkinson [9] & Patrick E. Savage [9,10] ✉

Music is a universal yet diverse cultural trait transmitted between generations. The extent to which global musical diversity traces cultural and demographic history, however, is unresolved. Using a global musical dataset of 5242 songs from 719 societies, we identify five axes of musical diversity and show that music contains geographical and historical structures analogous to linguistic and genetic diversity. After creating a matched dataset of musical, genetic, and linguistic data spanning 121 societies containing 981 songs, 1296 individual genetic profiles, and 121 languages, we show that global musical similarities are only weakly and inconsistently related to linguistic or genetic histories, with some regional exceptions such as within Southeast Asia and sub-Saharan Africa. Our results suggest that global musical traditions are largely distinct from some non-musical aspects of human history.

Do people and their cultures move together or independently? Darwin proposed "curious parallels"[1] between biological and cultural evolution, such that "a perfect pedigree of mankind... would afford the best classification of the various languages now spoken throughout the world"[2]. Darwin's proposal stimulated studies of cultural evolution that attempted to trace ancient population movements by combining linguistic, archaeological, and/or genetic histories[3–9]. Some have found support for a correspondence between the phylogenetic patterns in language and the movement of human populations[4,7,10,11]. For example, quantitative data comparing global genetic and linguistic diversity shows that genetic relationships between populations are generally tighter within language family groupings, but also that in around 20% of cases, populations are genetically closest to linguistically unrelated groups[12]. Critics of the phylogenetic approach argue for more complex relationships between people and their cultures, pointing out that basic

vocabularies used to construct language phylogenies represent one limited dimension of cultural history that does not necessarily correspond to other markers of language or culture[13–17].

Music, like language, is a universal cultural trait that varies within and between societies[18–24]. Could music play a significant part alongside language in research on human history? 70 years ago, Alan Lomax proposed that it could, arguing that musical style changes less than language or other cultural traits[25].

There are three contrasting predictions regarding potential historical relationships between music, languages, and genes: (1) music correlates with genes due to parallel processes of migration and evolution[26–29]; (2) music correlates with language due to shared process of cultural transmission via vocal and interactional domains (i.e., song and speech both primarily use words)[30–32]; (3) musical patterns are unrelated to either genes or language, due to differences in the evolutionary shape, fabric, and/or tempo of musical, linguistic, and

[1]Graduate School of Media and Governance, Keio University, Fujisawa, Japan. [2]Evolution of Cultural Diversity Initiative (ECDI), Australian National University, Canberra, Australia. [3]Association for Cultural Equity, New York, USA. [4]Department of Evolutionary Biology and Environmental Studies, University of Zurich, Zurich 8057, Switzerland. [5]Centre for the Interdisciplinary Study of Language Evolution (ISLE), University of Zurich, Zurich 8050, Switzerland. [6]Department of Life and Environmental Sciences, University of Cagliari, 09126 Cagliari, Italy. [7]Cohn Institute for the History and Philosophy of Science and Ideas, Tel Aviv University, Tel Aviv, Israel. [8]Edelstein Centre for the History and Philosophy of Science, Technology, and Medicine, Hebrew University of Jerusalem, Jerusalem, Israel. [9]School of Psychology, University of Auckland, Auckland, New Zealand. [10]Faculty of Environment and Information Studies, Keio University, Fujisawa, Japan. ✉e-mail: samuel.passmore@anu.edu.au; anna@culturalequity.org; patrick.savage@auckland.ac.nz

genetic evolution (e.g., rapid musical change independent of demographic or linguistic turnover)[33–35].

Direct quantitative testing of these competing predictions with matched musical, genetic, and linguistic data has previously been restricted to regional studies and has produced mixed results. Studies have found evidence of significant correlations between musical and genetic diversity in Taiwan, Sub-Saharan Africa, and Eurasia[26,36,37] but not in Northeast Asia[38]. Global comparisons were not previously possible because detailed public data on cross-cultural musical diversity, genetic similarity, and language history were not available. Recently, however, globally representative datasets of musical, genetic, and linguistic diversity have been published, allowing us to test these hypotheses.

Quantifying global musical diversity has been the focus of several recent efforts[22,24,39]. In this project, we use The Global Jukebox, a dataset of almost 6000 songs from almost 1000 societies coded on 37 standardized "Cantometric" features of musical style[20] (Table S1). The Global Jukebox is particularly well suited to comparing music, language, and genes across cultures. It contains over 15 times more coded songs than other datasets that use similar coding schemes[22,24]. While a slightly larger global dataset of 8200 audio recordings exists[18], it is constructed around the unit of country, which does not allow for direct comparison with genetic or linguistic data at the unit of ethnolinguistic groups. The dataset also relies on automated signal analysis and machine learning using country labels, without validating against human perceptual data. A recent independent examination of the data[40] found no significant correlation between the automated similarity algorithm[39] and human similarity ratings for a global sample of songs (but did find significant correlations between naive human similarity ratings and similarity metrics based on Cantometric codings), raising questions about the interpretability of automatically identified dimensions.

In this paper, we leverage the publication of the Global Jukebox with recently published datasets of genetic diversity, and global linguistic evolution[12,41,42], to directly compare these three domains on a global scale (Fig. 1). First, we use the musical data to extract five dimensions of musical style from the Cantometrics dataset and show that these dimensions contain between-group variability, making them useful markers for cultural history. Autocorrelational tests show that the between-group structure in our musical variables is organised geographically. We observe similar patterns in genetics and language, although the strength of the musical relationship is weaker than is found in genes or language. Finally, we show that the similarities among our five musical dimensions are only weakly related to the structure found in the linguistic and genetic data, such that musical traits capture largely independent information about human cultural history.

## Results

### Musical, linguistic, and genetic samples
Musical data is drawn from the Cantometrics dataset from within the Global Jukebox. Using this dataset, we devise three sample sets: a set where societies must have two or more songs to be included (resulting in a sample of 5242 songs from 719 ethnolinguistic groups), where societies must have 10 or more songs (3063 songs and 222 societies; Fig. S2), and a sample of societies matched to the Standard Cross-cultural sample (SCCS; 742 songs and 110 societies; Fig. S3).

Genetic data is drawn from GeLaTo, a genomic database designed to investigate patterns between genetic and linguistic diversity[12]. The dataset collects published genomic data genotyped with the Human Origins SNP chip, a chip designed to maximise human genetic variability across continents and minimise the effects of ascertainment bias[43], from 4000 individuals, representing 397 genetics populations and 295 languages. Most of the genetic data available, to our knowledge, was collected between the 1990's and today.

Linguistic relationships are drawn from a recently produced Bayesian global language phylogeny to quantify linguistic affiliations between the societies in our sample[41]. The global language phylogeny was built from a taxonomy of extant languages[44], together with previous Bayesian phylogenetic analyses of basic vocabulary data from major families, information on the timing of language diversification events, the geographic location of languages, and assumptions about the paths of human migration[41].

A detailed description of the processing steps of the Cantometrics dataset, the GeLaTo dataset, and the pairing of datasets can be found in the Methods and Supplementary Note 1. All statistical tests that follow are performed across the three Cantometric datasets to assess the robustness of effects, with the results found in the Supplementary Information, with a general summary in Tables S3 and S4.

### Five dimensions of Cantometric musical diversity
We first built a latent variable model containing five dimensions of musical style modelled after Lomax's[45] factor analysis of a subset of Cantometrics. Our five dimensions are a subset of Lomax's nine dimensions after excluding four dimensions due to coding interdependencies (See Table S6 and S7 for details). By using dimension reduction to reduce Cantometric variation to 5 latent musical variables, we distil any repeating signal that occurs across several interdependent variables, while removing information that is variable-specific.

The five latent dimensions were designed to reflect: (1) Articulation (lyric repetition and enunciation); (2) Tension (vocal width, nasality, raspiness); (3) Ornamentation (the amount of decorative singing within a song), (4) Rhythm (meter and tempo); and (5) Dynamics (volume, register, and intensity; see Supplementary Note 2 for more detail on variable construction and examples for all variables). To examine whether the five-dimensional model (including additional covariances; see Table S5) is a valid description of the data we use three tests of model validity: Root Mean Square Error of Approximation (RMSEA), Standardized Root Mean Square Residual (SRMR), and the Comparative Fit Index (CFI)[46]. Both RMSEA and SRMR values are considered appropriate model fits if they are <0.08. CFI is considered an appropriate fit if the score is above 0.9 (see Methods for descriptions of these measures).

The two or more song dataset (5242 songs, 719 groups) passes all model fit tests (RMSEA = 0.06 (90% CI: 0.056-0.068); SRMR = 0.05; CFI = 0.93). The model also fits a dataset where societies must have >10 songs to be included (RMSEA = 0.06 (90% CI: 0.059-0.061); SRMR = 0.06; CFI = 0.94) and a dataset that only contains societies within the Standard Cross-Cultural Sample (SCCS; RMSEA = 0.06 (90% CI: 0.059-0.067); SRMR = 0.05; CFI = 0.94). We performed additional sensitivity analyses on the latent variables by excluding all Cantometrics variables that show low inter-rater agreement (Cohen's kappa agreements of <0.4), proposed as a minimum acceptable level of reliability [e.g., in clinical contexts[47]]. All latent variables showed a correlation of 0.97 or higher between the two latent models, except Tension (Table S10). Results involving Tension should be interpreted cautiously. For simpler comparisons in later analyses, we separately build an aggregate measure of musical similarity over all Cantometric variables, analogous to the 'modal profiles' or 'musical distances' used in previous Cantometric analyses[20,23] (cf. Supplementary Note 3).

### Music varies between societies
Figure 2 shows that although musical diversity presents as a continuous phenomenon (Fig. 2a), there is also an underlying structure aligning with cultural lineages (Fig. 2b). Figure 2b shows overlapping yet different distributions of songs from two language families, Atlantic-Congo and Sino-Tibetan. Sino-Tibetan songs tend to be more ornamented, whereas Atlantic-Congo songs tend to contain more regular rhythms. Highlighting two societies within these language

families (Fig. 2b Red squares: Ubangi; Blue squares: Burmese) shows the multi-level nature of musical diversity within and between societies.

To formally test whether societies can be differentiated musically, we use an AMOVA (Analysis of Molecular Variance[23,48,49] test on a set of 636 societies with linked language families, totalling 5131 songs). AMOVA parses the variance of a trait to show the relative importance of within-society and between-society diversity (Supp. Data S1; cf. Fig. S5 for a visual comparison of results). In genetics, within-population variance accounts for 93–95% of variance and between-populations constitutes around 3% to 5% of the variance (Rosenberg et al.)[50]. Amongst our musical variables, within-society diversity explains between 54–72% of the total variation. Between-societies / within-macro group diversity (Macrogroupings can either be Language family or Macroarea) contains between 29–43%. Comparing musical and genetic fixation statistics (measures of similarity between populations)[51]; showed that the differences in music were between 10% and 40% higher than genetic differences between populations, although between-society musical diversity was also more variable (Fig. S6). In general, while musical diversity within societies is large, when compared to genetic populations there are substantial differences between societies. However, we caution that the relative between- vs. within-society variation is calculated slightly differently for music and genetics, and the greater diversity within music may partially reflect the diversity of different musical genres coexisting in each society's repertoire. More information on AMOVA analyses is held in Supplementary Note 3.

## Music is more similar between geographically closer societies

Societies have identifiable musical differences, but do these differences reflect geographic patterns as they do in genes and languages? We estimated geographic autocorrelation (literally self-correlation) by comparing the musical, linguistic, or genetic similarity between pairs of societies whose geographic separation falls within a specified distance class (e.g. all societies within 500 km), against the similarity of those outside (e.g. beyond 500 km). Distance between societies is calculated using Haversine distance (distance between two points on a sphere). Haversine distance is a crude but common distance metric for large-scale comparative studies[49,52] since it is the lowest assumption distance metric. A high correlation statistic indicates that the similarity between societies within the distance class is higher than those outside it, implying geographic autocorrelation. We calculate geographic autocorrelation for musical, linguistic, and genetic processes at 500 km intervals up to 20,000 km (Fig. 3).

Geographically closer societies are also more musically similar (Fig. 3 and Fig. S7–S9). Within the averaged measure of musical similarity, spatial autocorrelation persists up to 4000 km on average (p < 0.01), slightly less than the level of autocorrelation seen between societies when using linguistic or genetic distance, which persists to around 5000 km and 5500 km, respectively (Fig. 3). The average level of autocorrelation seen in music within 4000 km is 0.17, which is

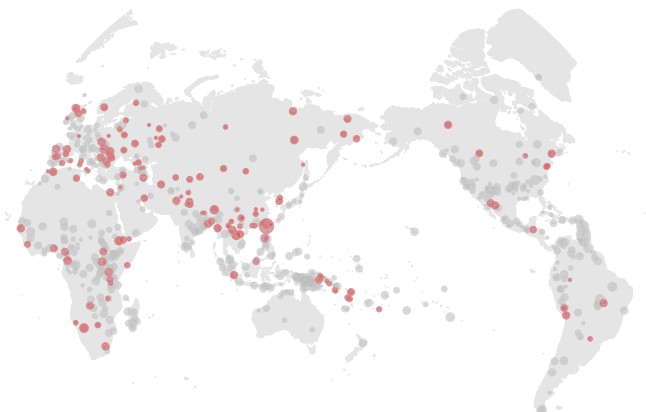

**Fig. 1 | Map of 719 Cantometrics societies (represented by 5242 songs).** Each point is a society, sized by the number of songs recorded for that society. All points are used to estimate latent variables. 121 societies (represented by 923 songs) are matched to both genetic and linguistic data and are coloured red. Societies without matching genetic and linguistic data are in grey. See Fig. S2 and S3 for maps of the 10 or more-song sample, and the SCCS sample. Maps made with Natural Earth.

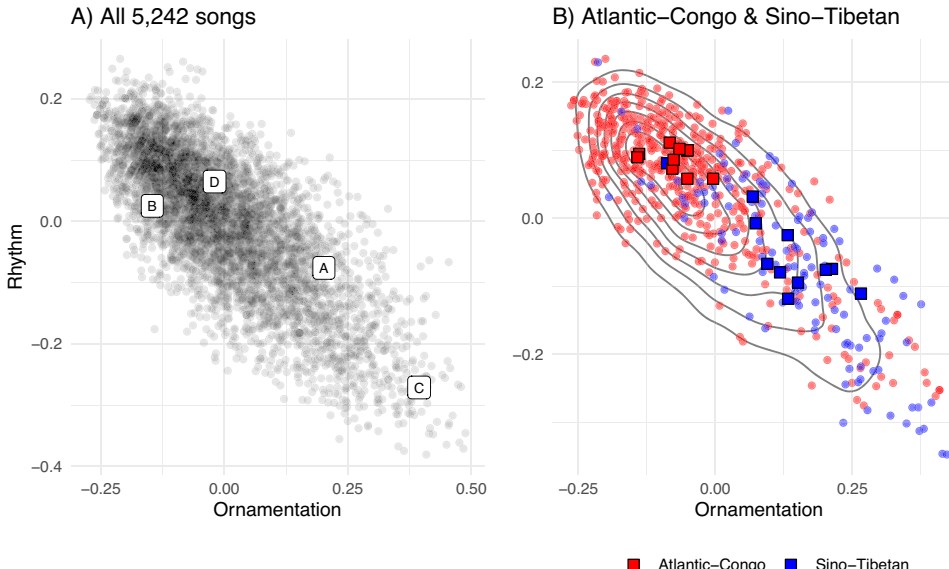

**Fig. 2 | Scatter plots between Ornamentation and Rhythm latent variables for 5242 songs. A** Each point is a song, and labels within the graph refer to exemplar songs described in S2.2: [A] Song with Xylophone—Burmese, [B] Djokobo—Mbendjele, [C] Caravan Song—Tibet, [D] Alima Song—Mbuti. **B** The same scatter plot as (**A**), with topographical gridlines showing the density of points in Fig. 2a, overlaid with points from the Atlantic-Congo language family (red) and the Sino-Tibetan language family (blue). Squares show songs from the Atlantic-Congo society Ubangi (red), and the Sino-Tibetan society Burmese (blue). The dispersal of squares reflects within-society diversity.

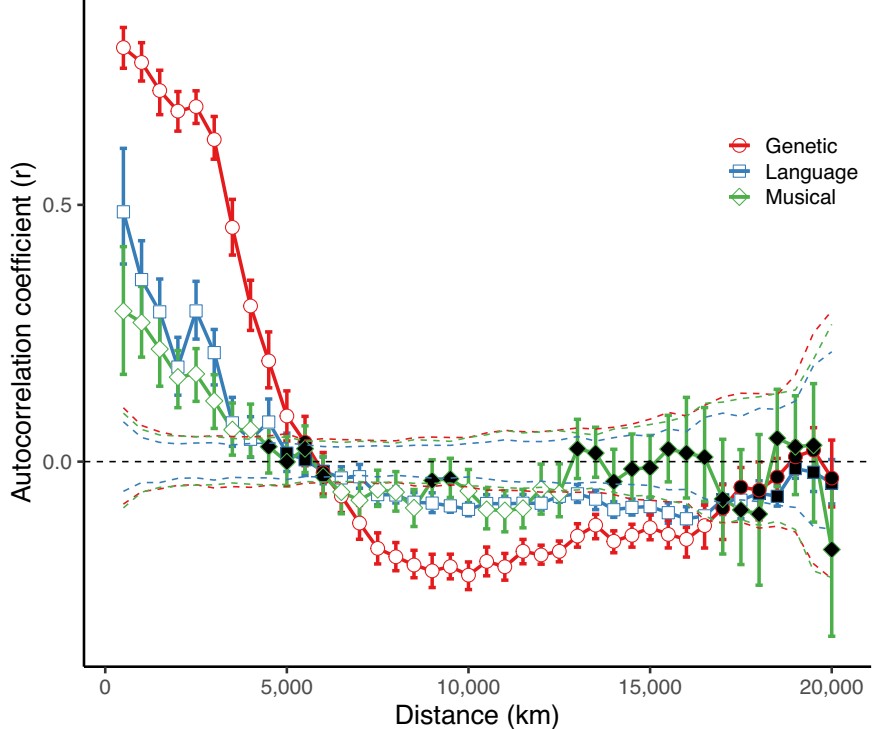

**Fig. 3 | A variogram showing the spatial autocorrelation coefficients (*r*) as a function of distance for society-level pairwise measures of Genetic F$_{ST}$ distances, phylogenetic distance from the global language phylogeny, and Musical Phi$_{ST}$ distances.** White shapes indicate significant autocorrelation and black shapes indicate non-significant autocorrelation. Error bars show the 95% confidence intervals for each distance. See Fig. S7 for the same graph for the individual musical metrics. See Fig. S8 and S9 for the 10 or more-song sample and the SCCS sample. See Supplementary Data 2–4 for detailed statistical information.

comparatively lower than the average value seen in language (*r* = 0.24) and genes (*r* = 0.63) across the same distance. This result is robustness tested for each of our three data samples, with statistics for each available Supplementary Data 2–4.

## Musical style suggests tree-like structure

Cultural evolution research often tries to separate the contribution of vertical and horizontal transmission. "Delta-scores" have been identified as a useful tool for identifying the extent of this conflicting signal[53–55], which quantifies how closely the distances between quartets of languages, music, etc., approximate the structure of a bifurcating tree (0 = a perfect branching tree with no reticulation, 1 = maximally non-tree-like). A perfectly branching tree represents a pattern of purely vertical transmission. Due to the computational expense of comparing all quartets, Delta scores are only calculated for 50 randomly chosen societies in each of Africa, Europe, and Oceania (the most sampled regions), for each latent variable. Delta scores range between 0.25 and 0.4 across all latent musical variables, with most results between 0.31–0.34 (Table 1). These values all fall between the

range previously reported in the lexicon of 12 Indo-European languages (0.21) and 38 Polynesian languages (0.41)[53]. They are also more tree-like on average than those reported for lexical and structural data in 81 Austronesian languages (0.38 and 0.44, respectively)[54].

## Musical similarity contains independent structure compared to language and genetics

We introduced three possibilities for why we might observe geographical patterning in musical style: (1) a correlation with genes; (2) a correlation with language; or (3) music is unrelated to genes or language. To dissect which of these theories is most likely we test the correlation between our musical measures, against measures of spatial, linguistic, and genetic similarity. We test the relationships between musical, spatial, linguistic, and genetic data using partial redundancy analysis (RDA)[38]. We also report analogous analyses using partial Mantel tests (Mantel, 1967), but caution that Mantel tests are often considered unreliable[56]. Musical distances are represented by Phi$_{ST}$ distance matrices (one for each dimension, and one for aggregate similarity, six in total). Genetic distances are measured by F$_{ST}$, and linguistic distances are measured through patristic distance.

Within our two or more song sample, music shows weak correlations with genes and language (Music−Genes (Controlling for Geography): Mantel's *r* = 0.15 (*p* < 0.001) RDA Adjusted *R*² = 0.09; Music−Genes (Language): *r* = 0.1 (*p* < 0.05); RDA Adj. *R*² = 0.06; Music−Language (Geography): *r* = 0.18 (*p* < 0.001); RDA Adj. *R*² = 0.1; Music−Language (Genes): *r* = 0.11 (*p* < 0.05), RDA Adj. *R*² = 0.03; See Supplementary Data S6 for more details). Results for the two or more-song sample show high agreement with the 10 or more-song sample (Supplementary Data S7), but not the SCCS sample (Supplementary Data S8; Table S12 summarises the comparison). The SCCS sample does not show strong relationships with any process. Of all 36 tests performed 91% returned an Adjusted R2 value of <10%, and 83% returned an

**Table 1 | Delta scores for 50 randomly chosen societies in Africa, Oceania, and Europe**

| Variable | Africa | Oceania | Europe |
|---|---|---|---|
| *Articulation* | 0.33 | 0.25 | 0.32 |
| *Dynamics* | 0.35 | 0.34 | 0.35 |
| *Ornamentation* | 0.34 | 0.33 | 0.29 |
| *Rhythm* | 0.33 | 0.31 | 0.33 |
| *Tension* | 0.33 | 0.35 | 0.33 |
| *All* | 0.38 | 0.40 | 0.36 |

The rows show the results for each latent variable.

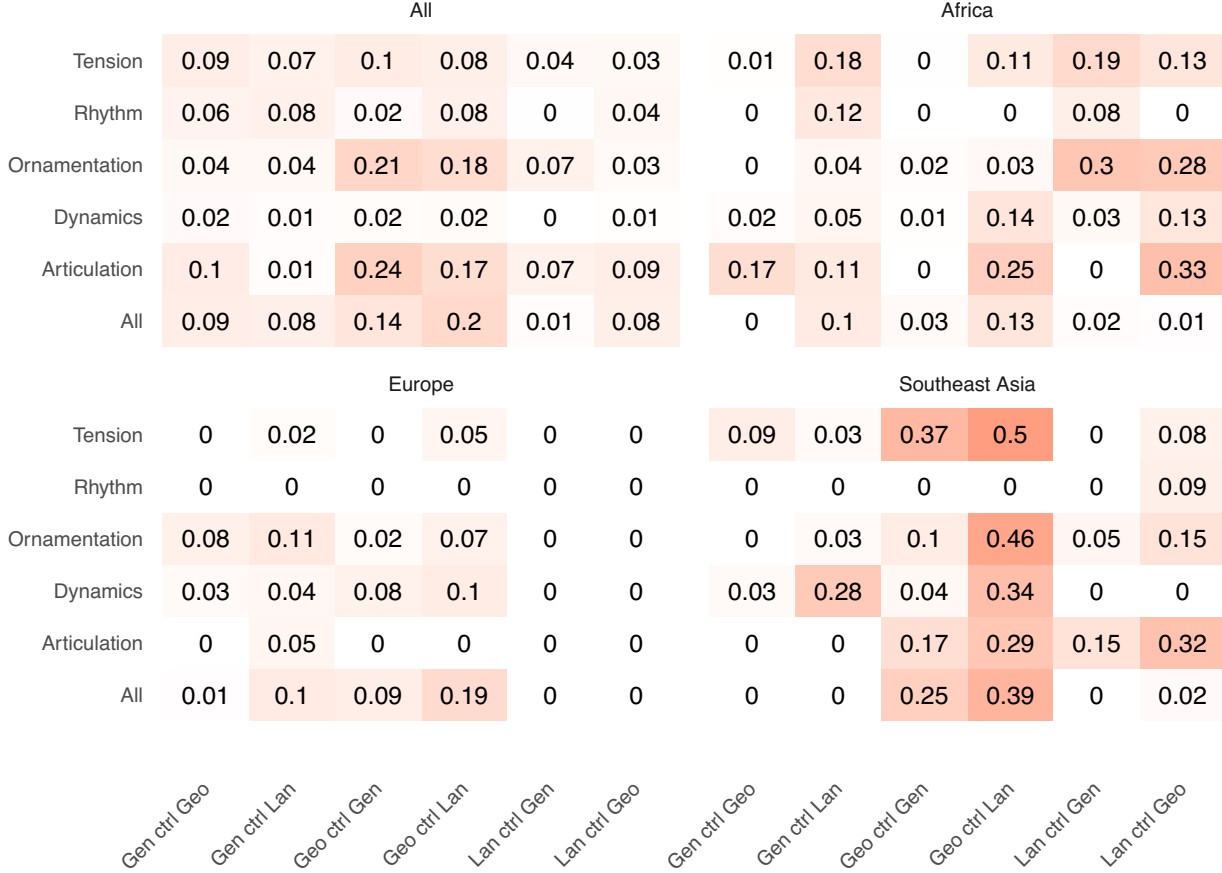

**Fig. 4 | Heat map of the Adjusted R² from partial RDA tests for each aspect of musical diversity.** RDA tests show the amount of variation explained by Genetic, Linguistic, or Spatial distances while controlling (ctrl) for a second process. From top-left, clockwise, Global distances, within Africa, within Europe, and within Southeast Asia. All correlations are rounded to two decimal places.

Adjusted R2 value of <5% (See Supplementary Data S8 for specific test results). Since the SCCS sample is designed to maximise the independence of societies (and thus minimize autocorrelation), we should expect that autocorrelation between groups is low in this dataset. To test the sensitivity of our results we also perform this analysis in regional samples.

Correlations within the three regions with the largest samples, Africa ($n = 20$ societies), Europe ($n = 27$), and Southeast Asia ($n = 12$), show us how variable the evolution of music may be, with the caveat of smaller samples (Fig. 4). Within Africa, the strongest musical correlations are with linguistic distance, explaining up to 33% of the variance (for the Articulation latent dimension). Within Europe and especially Southeast Asia, music is most correlated with geographic distances (up to 23% and 50%, respectively). We advise caution when interpreting the regional analyses for two reasons. Firstly, the smaller sample sizes in each region mean there is likely substantial unaccounted variability in the estimates, and secondly is variability in cultural homogeneity. A proxy for cultural homogeneity is the number of language families found within a region. In Europe and Southeast Asia, there are five language families each. Most European languages are Indo-European, with only a few Uralic, Turkic, and two isolate languages, whereas Southeast Asia contains a slightly less biased spread across the smaller families of Austroasiatic, Hmong-Mien, Sino-Tibetan, and Tai-Kadai languages, with the majority of languages coming from Austronesian. Within Africa, there are 22 different language families, considerably more than the other two regions. The number of languages in each family is heavily skewed towards a few large families, with 73% of samples coming from three language families. More than half of the

societies speak an Atlantic-Congo language, 15% are Afro-Asiatic, and 9% are Nilotic. In each regional case, there is a similar level of language family diversity, when accounting for their unbiased distribution. Nevertheless, the range and breadth of cultural diversity are likely to have an impact on regional calculations of similarity by acting as hurdles to the flow of genetic and cultural material.

## Discussion

After over a century of debate and partial testing using indirect proxies or regional samples, our direct comparison of global musical, linguistic, and genetic data reveals that musical histories—as captured in stylistic features of traditional songs—are not consistently related to genetic and linguistic histories on a global scale. This does not imply that musical features do not preserve historical patterns, as our analysis reveals that these features may preserve a relatively tree-like structure suggesting largely vertical transmission across generations. Rather, musical histories capture partially independent features from linguistic or genetic ones.

A practical concern might be that the three datasets are not comparable, and therefore a consistent relationship is not expected. We believe this is not the case for two reasons: (1) previous studies have shown that it is possible to capture significant correlations between cultural and/or genetic data using the same types of data sources used here[12,26,42]; and (2) our spatial autocorrelation analysis (Fig. 3) and regional analyses (Fig. 4) show the presence of stronger relationships in precisely the kind of local areas and scales that had previously been suggested in regional studies (i.e., sub-Saharan Africa and Southeast Asia[26,36]). This suggests that our finding of substantial

divergence on a global scale is not an artefact of our methodology but rather reflects the reality that music is largely independent of linguistic and genetic histories on a global scale. However, future analyses sampling music, language, and genes from the same population at the same time may be able to examine possible sampling biases more comprehensively (cf.[57,58] for examples of direct comparison of music and language recorded from the same individuals in diverse societies).

Our quantitative data seem to plausibly quantify what ethnomusicologists have long argued based on qualitative data: musical traditions often move independently from people or their languages. For example, the Cantometrics data has shown a large region of similar solo, heavily ornamented and richly accompanied singing styles stretching across the Eurasian "Silk Routes" from the Mediterranean to Japan, uniting groups speaking diverse languages from different families (e.g., Afro-Asiatic in North Africa, Indo-European, and Dravidian in India, Sino-Tibetan and Altaic in East Asia) and with partially independent genetic histories[42]. Instruments and musical systems have also been documented to have diffused and evolved across this trade route (e.g., the modern Japanese *shamisen* and European violin evolved from their shared ancestor with the Arabic *oud*[59]). Importantly, our results show that the topographical pattern of history that music holds is relatively independent of genes or language. While it may be the case that the rate of change is different between music, language, and certainly genes, the largely independent information we obtain from these three sources cannot be explained as the result of different rates of change. If traits changed at different speeds but primarily co-evolved through the same phylogenetic process, we would expect any correlation between co-evolving traits to be at least as strong as their correlation with geography. This is not what we find, suggesting that the patterns we observe reflect a pattern of global musical ancestry that is largely independent of the phylogenetic history of genes and languages.

There are many reasons why topologies might differ while maintaining historical structure. We propose two possible reasons why the historical topography of music differs from the other two phenomena. First, is the influence of historical patterns of borrowing. While languages are often used to delineate cultural groups, the historical frequency of bilingualism means individuals likely drifted between groups[60]. It is unclear if music adheres to the same level of boundedness, but it seems unlikely. Regional evidence discusses the exchange of musical ideas across linguistic and cultural boundaries[61]. The ability to share music across cultural boundaries means the flow of musical inheritance is not restricted to linguistic lineages. This might be classed as historical borrowing but could also be conceived as an alternative path of inheritance.

A crucial question for future research is to characterize the specific mechanisms driving the independence between music and language or genetics. Two major types of contrasting mechanisms are (1) neutral drift and (2) functional coevolution. Genetic and linguistic studies have primarily focused on markers that are not subject to strong selection pressure, also known as neutral markers (e.g., genetic variants unaffected by strong selective pressure, basic vocabulary for language) precisely because these are less likely to be influenced by functional coevolution[62]. Drift in the biological sense is impossible to realise in the absence of a genotype-phenotype division[63]. Within the study of music, it is not clear whether this distinction applies. In cultural-evolutionary studies in general, we rely on metaphor where drift is linked to usage-dependent mechanisms, the dynamics for which there is no purpose choice or benefit[64]. We know that some musical features have been proposed to evolve in a relatively neutral (no purpose) manner (e.g., microevolution of ornamental notes in melodies[65]), making drift a worthy hypothesis to test. Cantometric features were specifically designed to capture either functional coevolutionary relationships with social structure on a global scale or to track historical drift[20]. Modelling whether the hypotheses stand in the

light of the collected data will be a revealing avenue for evolutionary theories of music.

The current data alone will not allow us to differentiate between neutral and functional mechanisms. For example, virtuosic accompanied solo singing may have spread along the Bronze Age trade networks and the Silk Routes in a neutral manner accompanying trade and other cultural exchange, and/or functionally signalling and validating powerful hierarchies and the complex division of labour required to sustain large-scale trade networks[20,66] as has been proposed for other aspects of culture such as religion[67,68]. A reanalysis of proposed correlations between musical style and social structure supports Lomax's hypothesis of functional coevolution[42]. Future analyses directly comparing music, social structure, language, genes, and geography will be needed to explain causal mechanisms. Similarly, while our delta-score analysis suggests similar levels of horizontal and vertical transmission as found in previous analyses of language evolution (delta-scores ranging from roughly 0.2-0.4[53,54]), precise specification of these mechanisms of horizontal and vertical transmission in musical evolution will require more explicit models of the evolutionary process[30,69,70].

The global comparison of musical diversity to linguistic, and genetic diversity represents a substantial increase in size and geographic scope over previous regional analyses. Nevertheless, our data remain limited in important ways. In particular, the sample of 121 societies with matching genetic and linguistic data is only a small and non-random subset of the full musical sample of 719 societies (cf. Fig. 1), as available genetic data from Indigenous populations in the Americas, Africa, and Oceania is not well represented—often due to colonial legacies[71]. The full sample of Cantometrically coded musical data also only represents simplified reductions of the full complexity of cross-cultural musical diversity into 37 features, each of which has different levels of inter-rater reliability and cross-cultural universality (cf.[29,42,72] for critical discussion of the Cantometrics sample and methodology). While the reliability of the Cantometrics data has been validated using expert coders with substantial experience recording and analyzing music throughout the world[42], they have yet to be validated against the subjective judgments of culture-bearers who may not necessarily perceive their music in the same way as outsiders[73–75] (also see ref. 58 for an example of how perceptions of musicians themselves can be incorporated in comparative analysis). At the same time, it is equally possible that culture members find outsiders' observations interesting, useful, and validating when there is dialogue[76]. Similar limitations apply to the other datasets used for comparison: for example, linguistic phylogenies based on basic vocabulary data only capture certain aspects of language evolution[77] and biases in the design of genotype platforms can lead to skewed estimates of genetic diversity[78]. Our robustness analyses (Supplementary Data S7 and S8; Table S12; Fig. S11–S16) indicate that our current findings of broad independence between musical and linguistic/genetic histories are robust to specific sampling decisions regarding the populations or variables included in the current analyses. Future analyses comparing broader ranges of musical/linguistic features (e.g., grammatical features)[38,79]; direct acoustic comparison of sung/spoken audio[58,80] may help to understand the mechanisms underlying the separation.

The relative independence of musical processes in our analysis highlights the possibilities for music to tell us more about the relationships between societies. Earlier work has noted that "human history is written in both our genes and languages"[12] (page 1), but our work has shown that traces of history can also be found in other parts of culture. By expanding what we consider can tell us about human cultural history, we can build richer and more complex stories about the human cultural past, as well as the breadth of evidence used for building holistic models of human cultural history[12,81,82]. Much research on cultural evolution has shown complex connections between

cultural domains[83,84]. For example, that sex-biased movement creates distinct histories of language and material culture[85–89]. But it is equally possible for cultural domains to tell us about contrasting relationships in human history. Basketry traditions can transcend linguistic boundaries[90], and folk stories show incredible conservation across large geographical and historical areas[91,92]. Creative arts including music, dance, and poetry may be subject to less functional constraints and so may offer even more avenues in which culture can evolve independently of other aspects of cultural and population history[93–95]. Integrating models of music and the arts alongside genetic, linguistic, and other cultural histories into a unified narrative may enhance our knowledge of the shape and fabric of cultural evolution, and allow us to tell richer tales of the human past[53].

Ultimately, we show that relationships between musical styles are analogous to, yet largely distinct different from, linguistic and genetic relationships. Precisely how the interplay between neutral and functional mechanisms maintains musical similarity and drives musical change is still unclear. However, our publicly available data and code, combined with the recent release of complementary public datasets of global cultural and genetic data[12,42,79,96] provide an important foundation for future research into human cultural and biological evolution.

## Methods
### Data
The Global Jukebox contains Cantometric codings for 5776 songs from 1026 societies on 37 different variables[42] (Table S1, Fig. S1). The dataset used here was restricted societies with at minimum two songs, meaning we use a dataset with 5242 songs from 719 societies. We also restrict our analysis to 24 of the 37 variables, which are those without built-in redundancies (see Supplementary Note 2, Table S6, Table S7, Fig. S4). Songs can display multiple characteristics within a Cantometric Line throughout the performance, meaning some songs can have multiple codes for any particular variable. For analytical reasons, we require one value per song, per variable which we select at random. This affects 3% of the dataset. We built the latent variables 100 times with different randomly chosen values, finding that the average Pearson correlation between datasets was between 0.987 and 0.99.

All musical data is standardized to a 0–1 scale for comparability between features. We reverse the codes of several existing Cantometric variables so that all variables align high values with a more frequent occurrence of what the variable measures (Table S2). See supplementary material for more information on data pre-processing.

The Cantometrics dataset was designed and curated by Alan Lomax and Victor Grauer as an alternative to Western staff notation that could capture broad dimensions of musical performance present in varying degrees in the world's music, including not only rhythm and melody but also domains such as vocal timbre and social organization of the singers and accompanying instruments (Table S1). The coding scheme and first batch of several thousands of coded songs were first debuted in Lomax's landmark book Folk Song Style and Culture[20] and updated with thousands more songs over the subsequent decades. Most songs in the Global Jukebox database were coded by Lomax or Grauer themselves, and they have been recently validated for inter-rater reliability (mean $\kappa = 0.54$) and accuracy (-0.4–1% rate of unambiguous coding/data entry errors; Wood et al., 2022). The 37 variables vary substantially in reliability, from chance levels (e.g., nasality [Line 37]) to near-perfect (e.g., musical organization of the vocal part [Line 4] $\kappa = 0.94/89\%$ agreement[42]). Our robustness analyses removing low-reliability variables (Table S10) suggest that such variation does not affect our main results. Songs within the Global Jukebox were primarily collected from the 1940s to 1980s, with a maximum range of recording dates between 1904 and 1982.

In this paper, there are three sampling variants of the Canto-metrics dataset, that are used to test the robustness of the results: all societies that have two songs or more coded in Cantometrics, societies with 10 songs or more, and societies that align with the Standard Cross-Cultural Sample (SCCS).

The GeLaTo data consists of a sample of 1729 individuals from 156 populations, with a median of 9 individuals per population[12]. The source, glottocode and sample size of each genetic population sample are described in Supplementary Data S5 and can be found in a long format in the code repository. Genetic distances between populations are calculated with the Weir and Cockerham FST formula[97] implemented in the software PLINK v. 1.9[98], using the following script (https://github.com/epifaniarango/Fst_forLargeDatasets). The genetic distances are elaborated and expanded from a collection already described within the dataset GeLaTo[12]. In GeLaTo, published genetic data is merged, filtered and curated for anthropological and linguistic contextualization, to be used for multidisciplinary studies on human history and diversity. The genetic platform utilized in all the publications considered is the *Human Origins* SNP chip, a platform designed to maximize human genetic variability across continents and minimize the effects of ascertainment bias[43]. The final dataset includes individuals with a minimum of 550,000 SNPs successfully typed and calculates $F_{ST}$ over autosomal chromosome SNPs. We subset this matrix to 121 societies that could be paired to Cantometerics, totalling 923 songs. We convert the $F_{ST}$ distances to a correlation matrix using a Matérn correlation function, and the parameters: kappa = 0.001, phi = 0.1.

These musical, linguistic, and genetic datasets, like other cross-cultural datasets[96] as standardized ethnolinguistic markers to label and match societies, genetic populations and languages. We matched 121 societies from our musical dataset of two or more songs to the genetic and linguistic datasets (Fig. 1). 65 societies are paired via a direct match of Glottocodes across all datasets. A further 56 societies are manually matched with proxies, using higher- or lower-level glottocodes. For example, The linguistic phylogeny and genetic database contains samples for Czech (czec1258), but Cantometrics contains data for the subordinate dialect Czech-Morovian (czec1259). In this instance, the Cantometrics data is linked to the higher-level Czech data. The 121 societal matches are represented by 923 songs, the genomic profiles of 1296 individuals, and 121 languages across 38 language families. Our 10 or more song robustness sample is paired with 44 societies with linguistic and genetic data, and the SCCS sample to 21 societies (Fig. S2 and S3).

### Latent variable modelling
Latent variable modelling is performed using R v4.1[99] and the package lavaan v0.6-9[100]. In addition to the five latent variables, the model allows latent variables to correlate with each other and incorporates six correlations between Cantometric variables which were not explained by the latent variables. A written description of the latent variable model is given in Table S5. The coefficients for this model are standardized for both latent and observed variables, also known as a completely standardized solution. We use three common methods for the Latent Variable model Goodness of fit, RMSEA, SRMR, and CFI. RMSEA is the difference between the observed data variance (i.e. degrees of freedom) and the proposed model, penalizing for the number of parameters. A small value indicates the model explains close to the total variance in the data. A value of <0.08 is widely considered acceptable[46]. SRMR is a similar measure to RMSEA, but does not penalize for the number of parameters, again with values close to zero indicating better fit, and values <0.08 considered acceptable. These measures are called absolute measures of fit, and measure how far the model is from a perfect fit. CFI is a relative-fit measure, comparing the proposed model to a null model. A null model assumes all variables are independent. Values >0.9 are considered appropriate. The two or more song dataset (5242 songs, 719 groups) passes all model fit tests (RMSEA = 0.06 (90% CI: 0.056-0.068); SRMR = 0.05; CFI = 0.93). The model also fits a dataset where societies must have

>10 songs to be included (RMSEA = 0.06 (90% CI: 0.059-0.061); SRMR = 0.06; CFI = 0.94) and a dataset that only contains societies within the Standard Cross-Cultural Sample (SCCS; RMSEA = 0.06 (90% CI: 0.059-0.067); SRMR = 0.05; CFI = 0.94). The five dimensions align with Lomax's (1980) proposal of these five dimensions (a sixth dimension—organization—was not strongly supported and so was not included in analyses). Two of these dimensions—Ornamentation and Rhythm—share features with the two primary dimensions identified by Mehr et al. (e.g., their melodic complexity variable and our ornamentation variable both incorporate tremolo and melodic embellishment and their rhythmic complexity variable and our rhythm variable both incorporate tempo and metre). The other three dimensions—Articulation, Dynamics, and Tension—are not directly comparable because Mehr et al. did not include such features in their principal component analysis. Table S8 compares the weightings of a principal component analysis to the weightings of the latent variable analysis.

We tested the robustness of our results by running the analyses on the two or more-song sample, 10 or more-song sample, and the SCCS sample. Correlations between societies that exist between these datasets showed strong and significant correlations. All correlations are >0.97, with significant two-sided $p$-values (Table S9).

To ensure our results were not an artefact of coding bias, we performed additional sensitivity analyses on the creation of latent variables excluding all Cantometrics variables with Cohen's kappa agreements of <0.4, the lower end of the threshold described as moderate agreement[47]. One exception was made for Line 31 because of its importance in model convergence. In two cases this meant there was only one variable remaining in the latent variable model (Rhythm and Tension). Since we cannot create a latent variable from a single dimension, we use the single variables for comparison. We compare the full model to the remaining variable in these two instances. The comparison of the full to restricted latent variable model showed that all variables, except Tension, had significant Pearson correlations >0.7. Tension showed a significant, but small correlation to the remaining Tension variable (Table S10). Reported $p$-values are two-sided. Tension results then should be viewed with more caution than the other latent variables.

## AMOVA

AMOVA analysis is performed using R v4.1 and ade4 v1.7-18[101]. Information on Language family and the geographic Region categorisation are taken from Cantometrics metadata. Euclidean distances are calculated between songs. Macroareas are a geographical categorisation within Cantometrics that broadly correspond to United Nations Regional Groupings and Cantometric Regions and Divisions (https://unstats.un.org/sdgs/report/2019/regional-groups/). Results are available in Supplementary Data S1.

AMOVA analysis is additionally performed on societies with 10 or more songs, and the SCCS sample. The results show negligible differences between the two-song sample, and the 10 and SCCS samples (Supplementary Data S1, Fig. S5).

## Musical Phi$_{ST}$ and Genetic F$_{ST}$

Musical Phi$_{ST}$ matrices are created using the pairPhiST function within the haplotypes R Package[102]. We build Phi$_{ST}$ matrices for each musical dimension, and for an aggregate musical similarity that uses all Cantometric variables, for a total of five Phi$_{ST}$ matrices. Each of these five matrices is created three times, once for each sample of musical data. See the recipe Phi$_{ST}$ in the MakeFile for more details.

Genetic distances between populations are calculated with the Weir and Cockerham F$_{ST}$ formula[97] implemented in the software PLINK v. 1.9[98], using the following script (https://github.com/epifaniarango/Fst_forLargeDatasets). The genetic data comes from published sources that used the Human Origins SNP Chip, a panel that includes ~550,000 SNPs selected to be variable in populations from all continents[43].

F$_{ST}$ values are calculated from a sample of 121 populations, with a minimum of 5 individuals per population, a mean of 9, a maximum of 75, and a total of 1492 individuals.

## Spatial autocorrelation

Spatial autocorrelation for musical, and genetic variables were calculated from the distance matrices produced through the process in the previous section. A linguistic distance matrix was additionally produced using patristic distance within the global phylogeny, and a geographic distance was produced using Haversine distance, and the longitude and latitude for each society in the Cantometrics metadata (as described in the main text).

Autocorrelation was calculated using the Excel add-in Genalex[103]. We used the procedure "Single Pop Spatial Structure" found in the Distance-based menu, under the Spatial subheading. We used 40 evenly distributed spaced distance classes, which equated to 500 km bands, with a maximum distance of 20,000 km. These results are performed for each latent variable, the aggregated variable for the two-song dataset, and the aggregated musical variable in the 10 songs, and SCCS samples (Fig. S7–S9). We also produce a measure of autocorrelation for the genetic and linguistic data for all samples.

## Delta scores

Delta scores are based on measuring distance from the four-point condition[55]. The four-point condition says that, given four taxa have come from a tree (x, y, u, and v), the distances between those points must satisfy the following formula:

$$d(x, y) + d(u, v) <= \max\{d(x, u) + d(y, v), d(x, v) + d(y, u)\} \quad (1)$$

That is, the summed distance between x and y (x–y) and between u-v, must be less than the summed distance between x-u and y-v, or the summed distance between x-v and y-u, whichever is larger. Delta scores measure the distance from a perfect tree, meaning small scores are more treelike. We use Euclidean distance between each society's musical scores to calculate Delta scores.

We calculate Delta scores for samples of 50 societies in the two-song sample. The exponential increase in possible quartets with every increase in societies creates significant computational effort, and hence why we restrict the sample size. Societies are sampled at random from the two-song, 10-song, and SCCS datasets. The results for all these samples can be found in Table S11 and Fig. S10.

## Partial RDA and partial mantel

Musical Phi$_{ST}$ distance matrices were created using the function pairPhist in the haplotypes package (Aktas, 2020). The partial RDA analysis is a two-step process. First, we reduce all distance matrices to their primary dimensions using Principal Coordinate Analysis (PCoA). We extract all dimensions that explain >10% of the total variance. Then, we use RDA models to measure the correlation between the primary dimensions of the PCoA. Bi-variate RDA regresses a response variable set onto an explanatory variable set. Partial RDA allows us to assess the strength of correlation after controlling for the influence of a third confounder (e.g., regressing Articulation on genetic distance, controlling for geographic distance). We assess the strength of the relationship using adjusted R². Partial Mantel tests, like Partial RDA analyses, aim to estimate the correlation between two distance matrices while parsing the influence of a third. Partial Mantel tests were performed using mantel.partial in the vegan package[104]. Results for partial Mantel tests are in Supplementary Data S7. Mantel tests calculate the correlation between the two matrices, and then permute the rows of the response matrix to determine if the correlation is significantly greater than chance in a way that accounts for the non-independent nature of the distance matrix.

Partial Mantel and RDA tests are performed with the 10 or more-song sample and the SCCS sample. Results show a strong correlation between the 10 or more-song sample and the two or more-song sample, but neither sample showed a strong correlation to the SCCS sample (Table S12). The absence of a correlation with the SCCS sample is expected given that the SCCS sampling strategy intended to reduce autocorrelation in the data. All RDA $p$-values (Supplementary Data S6–S8) are two-sided tests.

## Changes to pre-registration

We registered a preliminary pre-registration of secondary data analysis, also available within the OSF archive. In the process of carrying out the analysis, our methods have changed substantially such that the current analyses should not be considered strictly pre-registered. The analyses here focus on RQ1 and RQ2 of the pre-registration, modelling the autocorrelational structure of the musical data. Our hypotheses of the major axis of musical style were expanded from three to five dimensions after realizing that our three originally proposed latent variables (social context, song structure, and singing style) were not the best variables to capture the primary dimensions of musical diversity (this change was done before analyzing correlations between these musical dimensions with genes, languages, or geography). The autocorrelational models proposed in RQ2 did not converge, therefore we shifted from a Bayesian to a frequentist modelling approach for the same hypotheses. We did not have time or space to sufficiently explore RQ3, which we will explore in future projects. The original pre-registration was registered on May 4, 2021, and can be found at https://doi.org/10.17605/OSF.IO/VE2DC.

## Inclusion and ethics statement

This study uses publicly available data[12,41] and so did not require additional ethical approval. For information about ethics and inclusion in primary data collection, please see refs. [12] and [41]—particularly the "Ethics, Rights, and Consent" and "Inclusivity in Global Research" sections. As noted there: "Repatriation of Lomax's recordings to their communities of origin, in partnership with those communities, is ongoing and has reached over 50 communities, descendants of artists, and national libraries. North American and Australian Indigenous audio samples will be streamed on the Jukebox only with the agreement of each community. To improve ethical practices, ACE [the Association for Cultural Equity] convenes with cultural advocates from diverse communities.... To further improve access to Lomax's recordings and research, ACE engages with community arts leaders, artists and other culture bearers to connect their constituencies to the Global Jukebox and our online archive in meaningful ways. They are invited to contribute Journeys and Exhibits, correct metadata, interpret the songs, suggest new songs and codings, and add their documentation to the songs." No identifiable information within the publicly available genetic dataset is available since all data is aggregated to the population level. Genetic data is used conforming to the associated informed consents and ethical permits, which allow the use of the data for studies of population history.

## Reporting summary

Further information on research design is available in the Nature Portfolio Reporting Summary linked to this article.

## Data availability

All data processed and used in this study are accessible at https://doi.org/10.5281/zenodo.10817212. All data is freely accessible. The processed data are available at the same address within the folder 'processed_data'. Some results and data processing take significant computing time, so we keep pre-computed results in the same repository and folder. This project only utilises existing datasets.

The sources of the data are as follows: The Global Jukebox[42], with data accessible from https://github.com/theglobaljukebox/cantometrics; GeLaTo[12], with data accessible from https://github.com/gelato-org/gelato-data, additionally, the source of the population samples used are also listed in Supplementary Data S5; The global language phylogeny[41], with data accessible at https://osf.io/yzxv9/. To listen to the audio, and read more detail on the Cantometric coding scheme visit http://theglobaljukebox.org. Please cite[42] if using Cantometrics, or other Global Jukebox data. Global Jukebox datasets are archived with ZENODO, and the DOI provided by ZENODO should be used when citing releases of Global Jukebox datasets, which are available within the GitHub organization.

## Code availability

All necessary code for replicating our analyses has been deposited into the ZENODO repository: https://doi.org/10.5281/zenodo.10817212. Each step of the analysis is detailed in chronological order within the Makefile.

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

## Acknowledgements

We thank all the individuals and researchers who previously contributed and curated the genetic, musical, and linguistic data. We thank Alan Lomax, Luca Cavalli-Sforza, Victor Grauer, Steven Brown, Sarah Tishkoff, Floyd Reed, and Armand Leroi for inspiration and discussion about comparing global patterns of musical and genetic diversity. We thank Russell Gray, Shinya Fujii, and members of the CompMusic Lab, NeuroMusic Lab, and Language, Culture, and Cognition Lab for feedback on earlier versions of the manuscript. The Global Jukebox has been developed with support from the National Endowment for the Arts, the National Endowment for the Humanities, the Concordia Foundation, the Rock Foundation, and Odyssey Productions. SP, HD, and PES are supported by funding from the Japan Society for the Promotion of Science (Grant-in-Aid #19KK0064); the Yamaha corporation; and grants from Keio University (Keio Global Research Institute and Keio Academic Development Fund). SP is also supported by the Evolution of Cultural Diversity Initiative at the Australian National University. PES is also supported by the Royal Society Te Apārangi (Rutherford Discovery Fellowship RDF-UOA2202 and Marsden Fast-Start Grant MFP-UOA2236). CB was supported by the University Research Priority Programme of Evolution in Action of the University of Zurich, the NCCR Evolving Language, the Swiss National Science Foundation Agreement (#51NF40_180888), and the SNSF

Sinergia project 'Out of Asia' (183578). Article Processing Charges are supported by the University of Auckland Faculty of Science Open Access High Impact Publication Fund. The funders had no role in study design, data collection and analysis, decision to publish, or preparation of the manuscript.

## Author contributions

PES, SP, and ALCW conceived the project. SP conducted the analyses with methodology recommended by PES, CB, QDA, ALCW, and DS. CB provided and curated the genetic data. SP, CB, DS, and HD curated, matched, and cleaned the data. HD conducted a code review. SP wrote the initial draft, with substantial revisions by PES and ALCW. CB, DS, and QDA contributed to the final draft.

## Competing interests

The authors declare no competing interests.
