## [Peer Review File · Nature Communications]

Global musical diversity is largely independent of linguistic and genetic historiesEditorial Note: This manuscript has been previously reviewed at another journal that is not operating a transparent peer review scheme. This document only contains reviewer comments and rebuttal letters for versions considered at *Nature Communications*. Mentions of the other journal have been redacted.

REVIEWER COMMENTS

Reviewer #1 (Remarks to the Author):

The authors present analyses that capitalize on previous large-scale efforts to collect data related to population genetics, languages, and musical practices to delineate relationships between diversity across these domains. Results are taken to suggest musical diversity across geographic regions is largely distinct from linguistic and genetic diversity, though some weak relationships were observed.

Overall, the study constitutes an impressive comparative effort of the sort that should be applauded. I have reviewed two previous versions at another journal and the authors have addressed all my previous comments. I am of the view that it is a fine piece of work that would be a welcome addition to the literature.

Reviewer #4 (Remarks to the Author):

Dear Editor,

Thank you for giving me a chance to review this excellent piece of research, and the peer-review exchanges around it, which I truly learned a lot from.

I start by briefly answering the focal question contained in your review invitation: no, I do not agree with Reviewer #3 that what they considered to be increased (relative to the initial version) doubts about the robustness of the findings are grounds for rejecting the paper at this stage. I find the authors' answers grounded in solid knowledge of statistics (suitably referenced), which in my personal view sufficiently rebuts the reviewer's (rather surprising) recommendation to reject.

In particular, it could be that, if available to respond, R#3 would fill in some missing information, e.g. what they consider to be a threshold for Cohen's kappa that is in their view more appropriate for comparative musicology studies. But absent such a follow-up exchange, I take the authors' side in their thorough (at times harsh) rebuttal of R#3's (statistical and other) arguments for rejection. I hazard a guess that it is not unthinkable for R#3 to have themselves been convinced by this very good rebuttal, had they been available for re-review.

The only thing I'd say re R#3's statistical criticisms is this. The crux of this paper is clearly the fact that music variables show (at least for Europe and SE Asia) a stronger relation to geographic distance than they do to variability in language and in genes. Given that the latter relations were however not quite null*, I just wonder whether a slight qualification of the word "independent" in the paper title (e.g. "largely independent", as used later in the Discussion, L318), would more accurately reflect the result of the spatial autocorrelation analysis and of the regional analyses. I again hazard a guess that this might well suffice to placate even the concerns of someone who (while accepting the authors' rebuttal), nonetheless may have some lingering doubt about the reliability of the datasets (cf e.g. the Cohen's kappa issue), or the robustness of the findings.

*: If I correctly understand the adjusted R² values from partial RDA tests in figure 4, these weren't all zero for the Gen (ctrl..) and Lan (ctrl..) columns, just on average smaller (relative to the appropriately-adjusted statistical threshold, of course) than the Geo (ctrl..) columns.

A more minor statistical point, which the authors offer to change, I personally (unlike R#3) do not find necessary to change, as I think it would just add one more conversion step in the (otherwise, perfectly justifiable) variable-rescaling pipeline, whilst in no way changing the results:

"We can technically remove the step of converting the data to sequential integers before converting to a 0-1 score. The only step that needs to occur is (e.g.) converting a fivecategory integer scale to a 0 – 1 scale. We are happy to make this simplification if it resolves the reviewer's concern. It will have no

impact on the results or conclusions, however."

The "disappointing" reduction in sample size to "only"(!) 121 societies is certainly no grounds for rejection, and constitutes in my view a simple case of an "anchor effect": while true that the starting number of societies in Cnatometrics was >1000, as the authors say, the remaining 121 (necessary if they were to pair the music data with the genetic and linguistic databases!) are still well more numerous than what previous studies had to work with.

In addition to interpreting Reviewer 3's concerns, I take, if I may, the chance in the remainder of this document (which I also attach a Word file) to make a few other comments of my own, as well as some minor suggestions, all of which might perhaps help the authors improve the paper further, ahead of the acceptance that I hope you will agree it deserves.

Once again thanks for the opportunity to review, and many congrats to the authors! Best wishes,
Tudor Popescu

1. COMMENTS ON THE MAIN RESULT

I wonder if the paper's main result can be interpreted to imply (in a simplified way) that people tend to copy the music of their neighbors, even if distantly "related" to them. If so, that would have the agreeable implication of supporting the intuitive notion of music as something that unites people(s) in spite of their differences. This is indeed what the authors also seem to suggest, in their discussion of the (wonderfully exciting) Silk Road result of Wood et al. 2022 (L323-332). On that, I'm not at all a population geneticist, but I wonder if we can truly be sure that those groups have had entirely "independent genetic histories" from one another, at least compared to groups outside of that Silk Road corridor? Anyway, this might only necessitate adding a slight nuance (if even) to an otherwise very well written paragraph.

As for the "how come?" question of the (causal) mechanisms underlying this difference, I wonder if, in addition to what the authors already eloquently speculate about in the discussion, this independence of music from language histories could also be related to, first, faster transmission time scales for music than for language (more transmission episodes per unit of time). And second, to neurocognitive mechanisms underlying each of those domains, which crystallise differently throughout development and across the lifespan. For instance, one typically (excluding bilingualism etc.) has a single "L1" as a mother tongue, but the same exclusivity and privileged neural "access" (as regards brain connectivity etc.) doesn't seem to be a requirement for the acquisition and enculturation into a musical tradition. Just a thought...

Another likely candidate mechanism that came to mind I encountered not in the paper, but only later when reading the rebuttal to Rev #3. This namely relates to patterns of borrowing, and I wonder whether the very clear explanation that the authors write in that rebuttal (below) wouldn't well deserve promotion to the main text of the paper:

"The most likely topological difference is likely to arise from patterns of borrowing. As we state in the text, there are no agreed "core musical traits" akin to the slow evolution of basic vocabulary. So, the prevalence and pattern of borrowing seems the most likely reason why we observe no relationships between musical and linguistic trait"

Still related to the idea of different transmission time scales is the following sentence at the end of the rebuttal, which the authors might wish to likewise consider perhaps finding a place for in the paper, as it once again (I think) speaks to the difference between "independent" and just "largely independent":
"We do not rule out [the possibility that music changes faster than basic vocabulary and genes], but it also does not preclude the possibility that music contains historical information"

Finally on the mechanisms, I'd have thought that the present data is in a position to tentatively lend

more support to neutral drift as a mechanism for the evolution of musical features, than to functional coevolution [at least wrt language or genes, if that counts as the "null" result here]. The authors argue it in fact does not allow differentiating between these two mechanisms, which I'm prepared to accept, with just one doubt. They refer to the example of virtuosic accompanied solo singing, developing contemporaneously with trade along various routes. Contrary to my previous thought of the data weighing in for neutral drift, I just wonder (for what it's worth) whether it is in fact even really plausible for this type of singing to have spread in a truly neutral manner as they speculate, i.e. with no function at all, such as - as the authors note - signaling and validating hierarchies, and division of labor arrangements. I guess my comment boils down to this: is it not always a question of finding the right type of functional mechanism, with neutral drift being just a null placeholder explanation while the appropriate data is yet to arrive? Or are there aspects of music for which a truly "neutral drift" mechanism is worthy of serious consideration?

To conclude here, I fully agree that, as the authors write in the Discussion, "A crucial question for future research is to characterize the specific mechanisms driving the independence between music and language or genetics"; and that "future analyses sampling music, language, and genes from the same population at the same time may be able to examine possible sampling biases more comprehensively". That would be a huge project, but one that I certainly see this team (perhaps in collaboration with Sam Mehr's and Reyna Gordon's group) being able to pull off!

2. UNIQUE VARIANCE FOR FORMS OF CULTURE OTHER THAN MUSIC, TOO?

The authors acknowledge that the paper's main result "does not imply that musical features do not preserve historical patterns, as our analysis reveals that these features may preserve a relatively tree-like structure suggesting largely vertical transmission across generations. Rather, it suggests that musical histories capture relatively independent features from linguistic or genetic ones".

I just wonder whether the same thing can hold for forms of culture other than music and language, such as religion, briefly mentioned later in the Discussion. That is, whether tree-like bifurcations may occur in those traditions also (insofar as the delta 0-1 range is truly respectively coterminous with wholly-vertical vs wholly-horizontal transmission!), but with those bifurcations describing a descent-with-modification process that is independent of other (coexisting) forms of culture. In other words, if those other cultural histories (or "cultural kinds" in the terminology of Wagner & Tomlinson, 2022), such as religious traditions, can, like musical histories, also capture features that are at least to some degree independent from histories which they otherwise appear to be inextricably intertwined with. If so, and if I may digress some more, one other question that comes to mind is: just how much _more_ unique variance (i.e. variance not necessarily related to genes) might there be to explain, in human culture at large, in addition to the salient cases of music and language considered here?

Finally, the authors write in their rebuttal that "if musical and linguistic processes evolve from the same phylogenetic process, we would expect to find that the traits are correlated – regardless of their rate of change (i.e. the rate of change does not affect correlations if the tree topology is the same)". As a non-expert in phylogenetics, I found that explanation quite helpful and wondered if it wouldn't also be worth considering finding a place for, in the paper or at least in the supplementary file.

3. MINOR COMMENTS

- Figure 3 would be a lot easier to read if different markers were used for the three conditions (data sets). The (presumably) confidence bands around $y=0$ are also not described in the figure caption, and are a bit hard to see with those colour choices.
- L360: "The global comparison of musical diversity to linguistic, and genetic diversity represents a substantial increase in size and geographic scope over previous regional analyses, our data remain limited in important ways." – I think maybe an "Although" is missing at the start of the sentence?
- L377: "..may well find outsider's observations interesting,.." – outsiders'?

Reviewer #5 (Remarks to the Author):

This is an extremely impressive and ambitious study of the evolution of indigenous music traditions at a global scale. It leverages a massive database of songs to explore the extent to which patterns of diversity in various dimensions of musical features correspond with linguistic, genetic and geographical relationships of 121 populations. The preparation and analyses of the data are diligent, sophisticated and carefully explained, with appropriate signposting to the Supplementary Information for further detail. The musical data are parsed and interrogated with extreme thoroughness, while hypotheses concerning the “shape and fabric” of their evolutionary histories are tested using a variety of methods, from AMOVAs and phylogenetic networks (delta scores) to spatial autocorrelation analysis, partial redundancy analysis and partial Mantel tests. Importantly, these various approaches are carefully integrated into a coherent analytical structure – this is not about “throwing the kitchen sink” but leaving no stone unturned.

The results are well presented and discussed with appropriate nuance and context. The headline findings are that while musical traditions exhibit evidence of phylogenetic and spatial structure, their transmission histories are relatively independent of genes and languages (albeit with exceptions for some features in some regions). As the largest-scale and most systematic study of its kind, this discovery has important implications for our understanding of the evolution of musical diversity, which has long been a topic of interest in comparative musicology. It is also highly relevant to broader debates about the extent to which human cultural, linguistic and genetic diversity have evolved through the same basic processes and pathways. As such the paper is likely to resonate with a wide and diverse audience, including ethnomusicologists, anthropologists, archaeologists, and researchers in cultural evolution and gene-culture co-evolution. The paper is therefore exceptionally well suited to a high-profile, interdisciplinary journal like *Nature Communications* and in my view certainly meets the journal’s criteria in terms of data quality, analytical rigour, originality and significance.

I note that the paper has been through multiple previous rounds of review at another [redacted] journal and although I was not previously involved in those, it seems clear that the authors have worked hard to address concerns raised in previous submissions. I don’t therefore have much to add in terms of further suggestions – by now it’ll be clear that I think this is a really excellent study! – other than a few minor points/queries which the authors may wish to address. These are offered in the spirit of intellectual interest and engagement with the paper, rather than as essential revisions for publication:

- Delta scores (lines 246-259): I wonder how sensitive the delta score is to the number of taxa and therefore how useful it is to compare scores derived from different datasets. In the studies they reference, there appears to be some relationship between the “tree-likeness” inferred from the delta score and the number of taxa. It’s also notable that the scores derived from the 50 societies are all quite close (falling within a tight range of 0.36-0.4 for all features), which seems slightly at odds with the regional differences reported by other analyses. This raised a slight doubt in my mind about how far the delta score is being driven more by the number of taxa and therefore how informative it really is. How different might the delta score be if they’d chosen, say 25 societies per region, or 100? It would be reassuring to know that the delta score is reasonably stable and not overly dependent on the choice of how many taxa to include.
- RDA tests (lines 287 – 293): Is it possible that the regional differences in correlations between music, language and geography are partly due to the relative diversity/homogeneity of the sample populations? For instance, are multiple language families represented in each region? This could be important given the effects of language barriers on gene and culture flow. I tried to look up this information in the SI but wasn’t able to locate it. Perhaps a brief sentence or two on the linguistic/cultural composition of the regional samples would be enough to address this.

- Discussion (especially lines 389 – 395): The point about integrating musical diversity with genetic and linguistic data to shed light on the shape and fabric of cultural evolution is well taken. Further connections could also be drawn with studies of material culture, folklore, etc. to develop a comprehensive and holistic view of human history and cultural evolution.

Response to the Reviewers for:

Global musical diversity is largely independent of linguistic and genetic histories.

Reviewer #1:

The authors present analyses that capitalize on previous large-scale efforts to collect data related to population genetics, languages, and musical practices to delineate relationships between diversity across these domains. Results are taken to suggest musical diversity across geographic regions is largely distinct from linguistic and genetic diversity, though some weak relationships were observed.

Overall, the study constitutes an impressive comparative effort of the sort that should be applauded. I have reviewed two previous versions at another journal and the authors have addressed all my previous comments. I am of the view that it is a fine piece of work that would be a welcome addition to the literature.

We thank the reviewer for their repeated commitment to reviewing this paper, and for their positive comments.

Reviewer #4

Thank you for giving me a chance to review this excellent piece of research, and the peer-review exchanges around it, which I truly learned a lot from.

I start by briefly answering the focal question contained in your review invitation: no, I do not agree with Reviewer #3 that what they considered to be increased (relative to the initial version) doubts about the robustness of the findings are grounds for rejecting the paper at this stage. I find the authors' answers grounded in solid knowledge of statistics (suitably referenced), which in my personal view sufficiently rebuts the reviewer's (rather surprising) recommendation to reject.

In particular, it could be that, if available to respond, R#3 would fill in some missing information, e.g. what they consider to be a threshold for Cohen's kappa that is in their view more appropriate for comparative musicology studies. But absent such a follow-up exchange, I take the authors' side in their thorough (at times harsh) rebuttal of R#3's (statistical and other) arguments for rejection. I hazard a guess that it is not unthinkable for R#3 to have themselves been convinced by this very good rebuttal, had they been available for re-review.

We thank the reviewer for their comments in support of our rebuttal to Reviewer #3.

The only thing I'd say re R#3's statistical criticisms is this. The crux of this paper is clearly the fact that music variables show (at least for Europe and SE Asia) a stronger relation to geographic distance than they do to variability in language and in genes. Given that the latter relations were however not quite null*, I just wonder whether a slight qualification of the word "independent" in the paper title (e.g. "largely independent", as used later in the Discussion, L318), would more accurately reflect the result of the spatial autocorrelation analysis and of the regional analyses. I again hazard a guess that this might well suffice to placate even the concerns of someone who (while

accepting the authors' rebuttal), nonetheless may have some lingering doubt about the reliability of the datasets (cf e.g. the Cohen's kappa issue), or the robustness of the findings.

We have changed the title of the paper to *Global musical diversity is largely independent of linguistic and genetic histories* since we think this better reflects the findings and topic of the paper.

*: If I correctly understand the adjusted R2 values from partial RDA tests in figure 4, these weren't all zero for the Gen (ctrl.) and Lan (ctrl.) columns, just on average smaller (relative to the appropriately-adjusted statistical threshold, of course) than the Geo (ctrl.) columns.

Correct, they are not all zero. Although, they are very small. We have added the following to the caption of Figure 4:

All correlations are rounded to two decimal places.

A more minor statistical point, which the authors offer to change, I personally (unlike R#3) do not find necessary to change, as I think it would just add one more conversion step in the (otherwise, perfectly justifiable) variable-rescaling pipeline, whilst in no way changing the results:

"We can technically remove the step of converting the data to sequential integers before converting to a 0-1 score. The only step that needs to occur is (e.g.) converting a five category integer scale to a 0 – 1 scale. We are happy to make this simplification if it resolves the reviewer's concern. It will have no impact on the results or conclusions, however."

We appreciate the reviewer raising this point. Since the reviewer agrees that this is unnecessary, we have not implemented this change in the coding scheme.

The "disappointing" reduction in sample size to "only"(!) 121 societies is certainly no grounds for rejection, and constitutes in my view a simple case of an "anchor effect": while true that the starting number of societies in Cantometrics was >1000, as the authors say, the remaining 121 (necessary if they were to pair the music data with the genetic and linguistic databases!) are still well more numerous than what previous studies had to work with.

We thank the reviewer for raising this point and wholeheartedly agree.

In addition to interpreting Reviewer 3's concerns, I take, if I may, the chance in the remainder of this document (which I also attach a Word file) to make a few other comments of my own, as well as some minor suggestions, all of which might perhaps help the authors improve the paper further, ahead of the acceptance that I hope you will agree it deserves.

Once again thanks for the opportunity to review, and many congrats to the authors! Best wishes,
Tudor Popescu

1. COMMENTS ON THE MAIN RESULT

I wonder if the paper's main result can be interpreted to imply (in a simplified way) that people tend to copy the music of their neighbors, even if distantly "related" to them. If so, that would have the agreeable implication of supporting the intuitive notion of music as something that unites people(s) in spite of their differences. This is indeed what the authors also seem to suggest, in their discussion of the (wonderfully exciting) Silk Road result of Wood et al. 2022 (L323-332). On that, I'm not at all a population geneticist, but I wonder if we can truly be sure that those groups have had entirely "independent genetic histories" from one another, at least compared to groups outside of that Silk

Road corridor? Anyway, this might only necessitate adding a slight nuance (if even) to an otherwise very well written paragraph.

We have added “partially” before “independent genetic histories” where this was mentioned (lines 269 and 287).

As for the "how come?" question of the (causal) mechanisms underlying this difference, I wonder if, in addition to what the authors already eloquently speculate about in the discussion, this independence of music from language histories could also be related to, first, faster transmission time scales for music than for language (more transmission episodes per unit of time).

Thank you for raising this point. It is an important distinction to make, which is also raised later in the review. Faster patterns of change, but the same topography would result in very strong correlations. What our results show is that the topography of musical, linguistic, and genetic histories, are likely to be different. We added the following text on line 291:

While it may be the case that the rate of change is different between music, language, and certainly genes, the largely independent information we obtain from these three sources cannot be explained as the result of different rates of change. If traits changed at different speeds but primarily co-evolved through the same phylogenetic process, we would expect any correlation between co-evolving traits to be at least as strong as their correlation with geography. This is not what we find, suggesting that the patterns we observe reflect a pattern of global musical ancestry that is largely independent of the phylogenetic history of genes and languages.

And second, to neurocognitive mechanisms underlying each of those domains, which crystallise differently throughout development and across the lifespan. For instance, one typically (excluding bilingualism etc.) has a single "L1" as a mother tongue, but the same exclusivity and privileged neural "access" (as regards brain connectivity etc.) doesn't seem to be a requirement for the acquisition and enculturation into a musical tradition. Just a thought...

Development is an interesting and important piece in our understanding of cultural change. However, we do not include a discussion on the role of development here. Although interesting, it would require significant space to address the idea appropriately, which we do not have. Since this paper does not consider developmental pathways otherwise, we believe the discussion around this mechanism is best had elsewhere.

Another likely candidate mechanism that came to mind I encountered not in the paper, but only later when reading the rebuttal to Rev #3. This namely relates to patterns of borrowing, and I wonder whether the very clear explanation that the authors write in that rebuttal (below) wouldn't well deserve promotion to the main text of the paper:

"The most likely topological difference is likely to arise from patterns of borrowing. As we state in the text, there are no agreed “core musical traits” akin to the slow evolution of basic vocabulary. So, the prevalence and pattern of borrowing seems the most likely reason why we observe no relationships between musical and linguistic trait"

We have added the following text to the discussion on line 296, in the spirit of this text:

There are many reasons why topologies might differ while maintaining historical structure. We propose two possible reasons why the historical topography of music differs from the other two phenomena. First, is the influence of historical patterns of borrowing. While

languages are often used to delineate cultural groups, the historical frequency of bilingualism means individuals likely drifted between groups (N. Evans, 2017). It is unclear if music adheres to the same level of boundedness, but it seems unlikely. Regional evidence discusses the exchange of musical ideas across linguistic and cultural boundaries (Daikoku et al., 2020). The ability to share music across cultural boundaries means the flow of musical inheritance is not restricted to linguistic lineages. This might be classed as historical borrowing but could also be conceived as an alternative path of inheritance.

Still related to the idea of different transmission time scales is the following sentence at the end of the rebuttal, which the authors might wish to likewise consider perhaps finding a place for in the paper, as it once again (I think) speaks to the difference between "independent" and just "largely independent":

"We do not rule out [the possibility that music changes faster than basic vocabulary and genes], but it also does not preclude the possibility that music contains historical information"

We have added the following sentence to the discussion on line 291, which is also mentioned above:

While it may be the case that the rate of change is different between music, language, and certainly genes, the largely independent information we obtain from these three sources cannot be explained as the result of different rates of change. If traits changed at different speeds but primarily co-evolved through the same phylogenetic process, we would expect any correlation between co-evolving traits to be at least as strong as their correlation with geography. This is not what we find, suggesting that the patterns we observe reflect a pattern of global musical ancestry that is largely independent of the phylogenetic history of genes and languages.

Finally on the mechanisms, I'd have thought that the present data is in a position to tentatively lend more support to neutral drift as a mechanism for the evolution of musical features, than to functional coevolution [at least wrt language or genes, if that counts as the "null" result here]. The authors argue it in fact does not allow differentiating between these two mechanisms, which I'm prepared to accept, with just one doubt. They refer to the example of virtuosic accompanied solo singing, developing contemporaneously with trade along various routes. Contrary to my previous thought of the data weighing in for neutral drift, I just wonder (for what it's worth) whether it is in fact even really plausible for this type of singing to have spread in a truly neutral manner as they speculate, i.e. with no function at all, such as - as the authors note - signaling and validating hierarchies, and division of labor arrangements. I guess my comment boils down to this: is it not always a question of finding the right type of functional mechanism, with neutral drift being just a null placeholder explanation while the appropriate data is yet to arrive? Or are there aspects of music for which a truly "neutral drift" mechanism is worthy of serious consideration?

We thank the reviewer for this thoughtful comment. As we state, the current data and analysis do not allow us to draw conclusions on drift vs functional mechanisms. But we can speculate on the issue. Part of the problem comes from how we use biological metaphors in cultural fields. True drift (in a biological sense) is impossible to envisage in the absence of a genotype-phenotype division. It is not clear at the moment how this distinction might be set up in a musical context. We are using a variation of the metaphor where drift is linked to usage-dependent mechanisms, the dynamics for which there is no purpose choice or benefit. In this sense drift (i.e. no purpose change) is a worthy hypothesis to test - even as only a viable null hypothesis.

We have added the following text to line 309 in the discussion to capture this idea:

Drift in the biological sense is impossible to realise in the absence of a genotype-phenotype division (Bickel et al. 2023). Within the study of music, it is not clear whether this distinction applies. In cultural-evolutionary studies in general, we rely on metaphor where drift is linked to usage-dependent mechanisms, the dynamics for which there is no purpose choice or benefit (Derex & Mesoudi, 2020). We know that some musical features have been proposed to evolve in a relatively neutral (no purpose) manner (e.g., microevolution of ornamental notes in melodies; (Savage et al., 2022)), making drift a worthy hypothesis to test. Cantometric features were specifically designed to capture either functional coevolutionary relationships with social structure on a global scale or to track historical drift (Lomax, 1968). Modelling whether the hypotheses stand in the light of the collected data will be a revealing avenue for evolutionary theories of music.

To conclude here, I fully agree that, as the authors write in the Discussion, "A crucial question for future research is to characterize the specific mechanisms driving the independence between music and language or genetics"; and that "future analyses sampling music, language, and genes from the same population at the same time may be able to examine possible sampling biases more comprehensively". That would be a huge project, but one that I certainly see this team (perhaps in collaboration with Sam Mehr's and Reyna Gordon's group) being able to pull off!

We thank the reviewer for their enthusiasm. A big task, indeed!

2. UNIQUE VARIANCE FOR FORMS OF CULTURE OTHER THAN MUSIC, TOO?

The authors acknowledge that the paper's main result "does not imply that musical features do not preserve historical patterns, as our analysis reveals that these features may preserve a relatively tree-like structure suggesting largely vertical transmission across generations. Rather, it suggests that musical histories capture relatively independent features from linguistic or genetic ones".

A note that we have softened the independence histories narrative to *largely* independent throughout the paper, in line with the reviewer's suggestion.

I just wonder whether the same thing can hold for forms of culture other than music and language, such as religion, briefly mentioned later in the Discussion. That is, whether tree-like bifurcations may occur in those traditions also (insofar as the delta 0-1 range is truly respectively coterminous with wholly-vertical vs wholly-horizontal transmission!?), but with those bifurcations describing a descent-with-modification process that is independent of other (coexisting) forms of culture. In other words, if those other cultural histories (or "cultural kinds" in the terminology of Wagner & Tomlinson, 2022), such as religious traditions, can, like musical histories, also capture features that are at least to some degree independent from histories which they otherwise appear to be inextricably intertwined with. If so, and if I may digress some more, one other question that comes to mind is: just how much *_more_* unique variance (i.e. variance not necessarily related to genes) might there be to explain, in human culture at large, in addition to the salient cases of music and language considered here?

This is precisely the idea we are trying to invoke, and we thank the reviewer for so eloquently discussing the idea. We added the following text to line 356 in the discussion which we think speaks to this idea:

By expanding what we consider can tell us about human cultural history, we can build richer and more complex stories about the human cultural past, as well as the breadth of evidence used for building holistic models of human cultural history (Aguirre-Fernández et al., 2021; Barbieri et al., 2022; Tambets et al., 2018). Much research on cultural evolution has shown

complex connections between cultural domains (Turchin et al. 2023; Henrich, 2020). For example, that sex-biased movement creates distinct histories of language and material culture (Buckley & Boudot, 2017; Lansing et al., 2017, Barbieri et al. 2013, Arias et al., 2018, Zhang et al. 2019). But it is equally possible for cultural domains to tell us about contrasting relationships in human history. Basketry traditions can transcend linguistic boundaries (Jordan & Shennan, 2003), and folk stories show incredible conservation across large geographical and historical areas (Tehrani, 2013, Bortolini et al. 2017). Creative arts including music, dance, and poetry may be subject to less functional constraints and so may offer even more avenues in which culture can evolve independently of other aspects of cultural and population history (Hoeschele & Fitch, 2022; Brown, 2021; Lomax, 1989). Integrating models of music and the arts alongside genetic, linguistic, and other cultural histories into a unified narrative may enhance our knowledge of the ‘shape and fabric’ of cultural evolution, and allow us to tell richer tales of the human past (Gray et al., 2010).

Finally, the authors write in their rebuttal that "if musical and linguistic processes evolve from the same phylogenetic process, we would expect to find that the traits are correlated – regardless of their rate of change (i.e. the rate of change does not affect correlations if the tree topology is the same)". As a non-expert in phylogenetics, I found that explanation quite helpful and wondered if it wouldn't also be worth considering finding a place for, in the paper or at least in the supplementary file.

Thank-you for this comment. We have added this sentence into the manuscript on line 292.

3. MINOR COMMENTS

Figure 3 would be a lot easier to read if different markers were used for the three conditions (data sets). The (presumably) confidence bands around $y=0$ are also not described in the figure caption, and are a bit hard to see with those colour choices.

L360: "The global comparison of musical diversity to linguistic, and genetic diversity represents a substantial increase in size and geographic scope over previous regional analyses, our data remain limited in important ways." – I think maybe an "Although" is missing at the start of the sentence?

L377: "...may well find outsider's observations interesting..." – outsiders'?

We have corrected all these minor suggestions.

Reviewer #5 (Remarks to the Author):

This is an extremely impressive and ambitious study of the evolution of indigenous music traditions at a global scale. It leverages a massive database of songs to explore the extent to which patterns of diversity in various dimensions of musical features correspond with linguistic, genetic and geographical relationships of 121 populations. The preparation and analyses of the data are diligent, sophisticated and carefully explained, with appropriate signposting to the Supplementary Information for further detail. The musical data are parsed and interrogated with extreme thoroughness, while hypotheses concerning the “shape and fabric” of their evolutionary histories are tested using a variety of methods, from AMOVAs and phylogenetic networks (delta scores) to spatial autocorrelation analysis, partial redundancy analysis and partial Mantel tests. Importantly, these various approaches are carefully integrated into a coherent analytical structure – this is not about “throwing the kitchen sink” but leaving no stone unturned.

The results are well presented and discussed with appropriate nuance and context. The headline findings are that while musical traditions exhibit evidence of phylogenetic and spatial structure, their transmission histories are relatively independent of genes and languages (albeit with exceptions for some features in some regions). As the largest-scale and most systematic study of its kind, this

discovery has important implications for our understanding of the evolution of musical diversity, which has long been a topic of interest in comparative musicology. It is also highly relevant to broader debates about the extent to which human cultural, linguistic and genetic diversity have evolved through the same basic processes and pathways. As such the paper is likely to resonate with a wide and diverse audience, including ethnomusicologists, anthropologists, archaeologists, and researchers in cultural evolution and gene-culture co-evolution. The paper is therefore exceptionally well suited to a high-profile, interdisciplinary journal like Nature Communications and in my view certainly meets the journal's criteria in terms of data quality, analytical rigour, originality and significance.

We thank the reviewer for their kind words, and are glad they appreciate the thoroughness of our analyses.

I note that the paper has been through multiple previous rounds of review at another [redacted] journal and although I was not previously involved in those, it seems clear that the authors have worked hard to address concerns raised in previous submissions. I don't therefore have much to add in terms of further suggestions – by now it'll be clear that I think this is a really excellent study! – other than a few minor points/queries which the authors may wish to address. These are offered in the spirit of intellectual interest and engagement with the paper, rather than as essential revisions for publication:

Delta scores (lines 246-259): I wonder how sensitive the delta score is to the number of taxa and therefore how useful it is to compare scores derived from different datasets. In the studies they reference, there appears to be some relationship between the “tree-likeness” inferred from the delta score and the number of taxa. It's also notable that the scores derived from the 50 societies are all quite close (falling within a tight range of 0.36-0.4 for all features), which seems slightly at odds with the regional differences reported by other analyses. This raised a slight doubt in my mind about how far the delta score is being driven more by the number of taxa and therefore how informative it really is. How different might the delta score be if they'd chosen, say 25 societies per region, or 100? It would be reassuring to know that the delta score is reasonably stable and not overly dependent on the choice of how many taxa to include.

We have improved the implementation of this algorithm and re-ran the statistic for a maximum of 100 societies per region for each of the musical dimensions. The computing constraints meant we still could not calculate the statistic for *all* variables. The results are now as follows:

Table 1: Delta scores for 100 randomly chosen societies in Africa, Oceania, and Europe. The rows show the results for each latent variable. When analysing all variables we only include 50 societies due to computational constraints.

Variable	Africa	Oceania	Europe
Articulation	0.30	0.27	0.32
Dynamics	0.31	0.34	0.34
Ornamentation	0.33	0.33	0.29
Rhythm	0.33	0.31	0.32
Tension	0.30	0.36	0.33
All (n = 50)	0.38	0.40	0.36

For convenience we have included the difference between the 50 and 100 sample analyses, although this is not included in the manuscript.

Table R1: Delta-score estimates from a 50 sample analysis minus 100 sample analysis

Variable	Africa	Oceania	Europe
Articulation	-0.03	-0.02	0.002
Dynamics	0.04	0.001	0.01
Ornamentation	0.004	0.002	-0.001
Rhythm	-0.01	0.004	0.01
Tension	0.03	0.01	0.001
All	-	-	-

As these results are not impacting the delta score, we are not further discussing this sensitivity test in the manuscript.

RDA tests (lines 287 – 293): Is it possible that the regional differences in correlations between music, language and geography are partly due to the relative diversity/homogeneity of the sample populations? For instance, are multiple language families represented in each region? This could be important given the effects of language barriers on gene and culture flow. I tried to look up this information in the SI but wasn't able to locate it. Perhaps a brief sentence or two on the linguistic/cultural composition of the regional samples would be enough to address this.

Thank you for raising an interesting point. We have added the following to line 244 in the Results section:

We advise caution when interpreting the regional analyses for two reasons. Firstly, the smaller sample sizes in each region mean there is likely substantial unaccounted variability in the estimates, and secondly is variability in cultural homogeneity. A proxy for cultural homogeneity is the number of language families found within a region. In Europe and Southeast Asia, there are five language families each. Most European languages are Indo-European, with only a few Uralic, Turkic, and two isolate languages, whereas Southeast Asia contains a slightly less biased spread across the smaller families of Austroasiatic, Hmong-Mien, Sino-Tibetan, and Tai-Kadai languages, with the majority of languages coming from Austronesian. Within Africa, there are 22 different language families, considerably more than the other two regions. The number of languages in each family is heavily skewed towards a few large families, with 73% of samples coming from three language families. More than half of the societies speak an Atlantic-Congo language, 15%

are Afro-Asiatic, and 9% are Nilotic. In each regional case, there is a similar level of language family diversity, when accounting for their unbiased distribution. Nevertheless, the range and breadth of cultural diversity is likely to have an impact on regional calculations of similarity by acting as hurdles to the flow of genetic and cultural material.

Discussion (especially lines 389 – 395): The point about integrating musical diversity with genetic and linguistic data to shed light on the shape and fabric of cultural evolution is well taken. Further connections could also be drawn with studies of material culture, folklore, etc. to develop a comprehensive and holistic view of human history and cultural evolution.

We agree that the discussion of quantitative models cultural evolution can be much broader than language. We have amended the penultimate paragraph of our paper (starting line 353) to point to some key papers in this area:

By expanding what we consider can tell us about human cultural history, we can build richer and more complex stories about the human cultural past, as well as the breadth of evidence used for building holistic models of human cultural history (Aguirre-Fernández et al., 2021; Barbieri et al., 2022; Tambets et al., 2018). Much research on cultural evolution has shown complex connections between cultural domains (Turchin et al. 2023; Henrich, 2020). For example, that sex-biased movement creates distinct histories of language and material culture (Buckley & Boudot, 2017; Lansing et al., 2017, Barbieri et al. 2013, Arias et al., 2018, Zhang et al. 2019). But it is equally possible for cultural domains to tell us about contrasting relationships in human history. Basketry traditions can transcend linguistic boundaries (Jordan & Shennan, 2003), and folk stories show incredible conservation across large geographical and historical areas (Tehrani, 2013, Bortolini et al. 2017). Creative arts including music, dance, and poetry may be subject to less functional constraints and so may offer even more avenues in which culture can evolve independently of other aspects of cultural and population history (Hoeschele & Fitch, 2022; Brown, 2021; Lomax, 1989). Integrating models of music and the arts alongside genetic, linguistic, and other cultural histories into a unified narrative may enhance our knowledge of the ‘shape and fabric’ of cultural evolution, and allow us to tell richer tales of the human past (Gray et al., 2010).

REVIEWERS' COMMENTS

Reviewer #4 (Remarks to the Author):

I thank the authors for having thoroughly addressed my comments. I am satisfied with the present revision and wholeheartedly recommend publication.

Reviewer #5 (Remarks to the Author):

The authors have fully addressed all of the comments from my previous review. I am pleased to recommend the paper for publication and believe it will make a valuable contribution to the literature.